# Design and Optimization of A Magneto-Plasmonic Sandwich Biosensor for Integration within Microfluidic Devices

**DOI:** 10.3390/bios12100799

**Published:** 2022-09-27

**Authors:** Mona Soroush, Walid Ait Mammar, Axel Wilson, Hedayatollah Ghourchian, Michèle Salmain, Souhir Boujday

**Affiliations:** 1Sorbonne Université, CNRS, Laboratoire de Réactivité de Surface (LRS), F-75005 Paris, France; 2Laboratory of Bioanalysis, Institute of Biochemistry & Biophysics, University of Tehran, P.O. Box 13145-1365, Tehran 1417614335, Iran; 3Sorbonne Université, CNRS, Institut Parisien de Chimie Moléculaire (IPCM), F-75005 Paris, France

**Keywords:** gold nanoparticles, magnetic beads, immunosensor, optical transduction, microfluidic chip

## Abstract

We designed a magneto-plasmonic biosensor for the immunodetection of antigens in minute sample volume. Both spherical gold nanoparticles (AuNP) and magnetic beads (MB) were conjugated to goat anti-rabbit IgG antibody (Ab) capable of recognizing a model target, rabbit IgG (rIgG). The AuNP bioconjugate was used as the optical detection probe while the MB one was used as the capture probe. Addition of the target analyte followed by detection probe resulted in the formation of a sandwich immunocomplex which was separated from the unbound AuNP-Ab conjugate by application of an external magnetic field. The readout was executed either in a direct or in indirect way by measuring the UV–Visible spectrum of each fraction in a specially designed microcell. Dose–response curves were established from the optical signal of the immunocomplex and unbound AuNP-Ab conjugate fractions. Finally, the assay was transposed to a microfluidic cell specially designed to enable easy separation of the immunocomplex and AuNP-Ab conjugate fractions and subsequent analysis of the latter fraction and achieve the quantification of the analyte in the ng/mL concentration range.

## 1. Introduction

Biosensors are analytical devices that find increasing application in medicine [1], food control [2] and environmental monitoring [3]. In its simplest configuration, a biosensor combines a bioreceptor acting as the recognition element that is typically immobilized on a solid support, called the transducer, that converts the recognition event into a signal whose magnitude can be correlated to the concentration of analyte.

Gold nanoparticles (AuNP) have been employed in a large variety of biosensing schemes [4], either as plasmonic transducers [5] or as optical [6] and non-optical [7] labels, owing to their versatile physico-chemical properties. One of the most remarkable features of AuNP is certainly their optical properties owing to the Localized Surface Plasmon Resonance (LSPR) effect that gives rise to an extinction band in the visible spectral range, with a typical maximum at 520 nm for 13-nm diameter spherical nanoparticles and extinction coefficients in the 10^8^–10^9^ M^−1^·cm^−1^ range. This remarkable feature enables their detection at very low concentration with a benchtop UV–visible spectrometer or even by the naked eye [8,9]. The position of the LSPR band of gold nanoparticles is also sensitive to minor changes of the local refractive index allowing optical transduction of target recognition in their vicinity [10].

On the other hand, magnetic beads (MB) enable the easy capture and (pre)concentration of analytes from complex media. They are commercially available with various surface functional groups, enabling facile immobilization of bioreceptors. Their high surface-to-volume ratio accelerates the rate of binding events compared to 2D (planar) materials [11]. Furthermore, they show an excellent compatibility with microfluidic systems in which they can be moved from one compartment to another by applying a magnetic field [12].

Several immunosensor configurations have been built up by association of MB and AuNP, taking advantage of the respective properties of both nanomaterials. The most widespread assay configuration makes use of the ability of AuNP to amplify the Raman signal of organic tags to build up immunosensors with a SERS readout [13,14,15,16,17,18,19,20]. Other strategies rely on the intrinsic optical properties of AuNP. For instance, the high refractive index of MB was exploited to set up a sensitive double-bead sandwich assay of cardiac troponin I with LSPR detection of the immunocomplex where AuNR acts as the transducer and MP as the signal amplifier [21]. The high extinction coefficient of AuNP was used to set up a sandwich assay of the prostate cancer biomarker PSA relying on MB-mAb conjugate capture probe (mAb = monoclonal antibody) and AuNP-mAb conjugate detection probe with readout by UV–vis spectrometry [22]. A similar approach was also reported for the competitive assay of the mycotoxin aflatoxin B1 [23,24] and biotin [25]. Alternatively, the strong Resonance Rayleigh Scattering (RRS) of AuNR was used to design a double-bead sandwich assay of α-fetoprotein with RRS detection [26]. The strong light-scattering properties of AuNR were also used to set up a sandwich assay of PSA combining MP-antibody and AuNR-aptamer conjugates with detection by dark field microscopy [27].

Herein, we report the design of a magneto-plasmonic biosensor for the detection of a model antigen, namely rabbit IgG (rIgG) in minute sample volume. Both spherical AuNP and magnetic beads were conjugated to goat anti-rIgG antibody (Ab). The former was used as optical detection probe and the latter as capture probe. Addition of the target antigen followed by detection probe resulted in the formation of a sandwich immunocomplex. Application of an external magnetic field enabled the separation of the immunocomplex from the unbound AuNP-Ab conjugate which were separately analyzed in a specially designed microcell. Dose–response curves were established from the LSPR signal of the immunocomplex and unbound AuNP-Ab conjugate fractions. Finally, the assay was transposed to a microfluidic cell specially designed to enable easy separation of the immunocomplex and AuNP-Ab conjugate fractions and subsequent analysis of the latter fraction.

## 2. Materials and Methods

### 2.1. Reagents and Buffers

Gold (III) chloride trihydrate (99.9%), sodium citrate dihydrate (99%), tannic acid (99%), 2-iminothiolane hydrochloride (Traut’s reagent), bovine serum albumin (BSA), rabbit IgG (I-5006), goat anti-rabbit IgG (R-5001), and fluorescein-5-isothiocyanate (FITC) were purchased from Sigma-Aldrich (St. Quentin Fallavier, France). Milli-Q water (18 MΩ·cm; Millipore, Molsheim, France) was used to prepare aqueous solutions. Experiments were carried out at room temperature unless specified otherwise. FITC-labeled goat anti-rIgG was prepared according to the literature [28].

### 2.2. Microcell for Absorption Measurements

The microscale cells (pathlength 1 mm; fill volume 70 µL) were designed with SolidWorks CAD software and printed by fused deposition modeling (FDM) using an Ultimaker 3 in tough polylactic acid (t-PLA). For this work, all the 3D models were printed using the following main specifications: layer thickness 100 µm, infill 70%, wall thickness six layers which resulted in good quality prints.

### 2.3. Microfluidic Cell

The microfluidic cell was fabricated using polydimethylsiloxane (PDMS) shaped by a 3D-printed mold. The mold was designed with Solidworks (Dassault Systèmes, Paris, France) and printed with a resolution of 25 µm using a laser stereolithographic (SLA) 3D printer (Form3, Formlabs Inc., Somerville, MA, USA) using their proprietary transparent resin (Clear V4, Formlabs Inc., Somerville, MA, USA).

The PDMS was mixed with a curing agent in 10:1 weight ratio, respectively. The mixture was sonicated for 10 min and then kept in a vacuum desiccator to remove all bubbles. A few mm thick layer of PDMS was poured into a flat petri dish, covered with the mold and subsequently heated at 80 °C for 2 h. The mold was then peeled, and holes punched for fitting PTFE tubing. The cell was completed by forming a glass-PDMS-glass assembly. A 150 µm thick standard 24 × 60 mm^2^ glass slide was used to cover the open channel of the cell, and a standard 150− µm thick, 18 × 18 mm^2^ glass cover slip was used to close the readout widow. Glass parts were glued to PDMS by placing the separated components in an oxygen plasma (45 s, 40 Pa, 1.2 L/min, 30 W; Harrick Plasma Inc., Ithaca, NY, USA) and subsequently heating the assembly at 80 °C overnight. The glass-PDMS-glass assembly was finally mounted on 3D-printed cartridge support, whose dimensions fit the Insplorion instrument.

### 2.4. Conjugation of Goat Anti-rIgG Antibody to Magnetic Beads

Commercial NHS-functionalized superparamagnetic beads (Pierce™ NHS-activated magnetic beads, ref 88826, ThermoFisher Sci., Les Ulis, France) were coupled to goat anti-rIgG according to the manufacturer’s instructions. In brief, magnetic beads’ slurry (150 μL) was pipetted into a 1.5 mL microtube. Using a magnet stand, magnetic beads were collected and the supernatant discarded. Then, ice-cold 1 mM hydrochloric acid (0.5 mL) was added and gently mixed. The magnetic beads were collected and the supernatant discarded. Goat anti-rabbit IgG solution (1 mg/mL in 50 mM borate pH 8.5; 150 μL) was added and incubated for 2 h under rotation. Magnetic beads were collected and 0.1 M glycine (pH 2.0; 0.5 mL) was added and mixed well. The magnetic beads were collected and the supernatant discarded. Then, 3 M ethanolamine (pH 9.0; 0.5 mL) was added, mixed well and incubated for 2 h under rotation. Magnetic beads were washed once with deionized water (0.5 mL) and twice with borate buffer (0.5 mL). Finally, MB-Ab conjugate was stored at 4 °C in 50 mM borate pH 8.5 (150 μL; final concentration 10 mg/mL) until use.

### 2.5. ELISA Test for Validation of Antibody Attachment to Magnetic Beads

Various concentrations of MB-Ab (125, 250, 375, 500 µg/mL) were dispensed in wells of a microplate. As a negative control, various concentrations of MB-BSA (125, 250, 375, 500 µg/mL) were dispensed in other wells. Using a magnet stand, the magnetic beads were collected and the supernatant discarded. HRP-labeled donkey anti-goat IgG in PBS-0.1% BSA was dispensed in the wells (1/8000; 200 µL). After 1 h incubation, the magnet stand was applied to keep the magnetic beads at the bottom of wells and the supernatant discarded. MB were washed 4 times with PBS-0.05% Tween 20 and freshly prepared o-phenylenediamine dihydrochloride (OPD) and the H_2_O_2_ mixture (7 mg OPD + 4 μL H_2_O_2_ in 10 mL citrate-phosphate buffer pH 5; 200 μL) was added to each well. After color development, 2.5 M H_2_SO_4_ (50 µL) was added to stop the reaction, yellow-colored supernatants were transferred to empty wells and OD485 nm read with a microplate reader.

### 2.6. AuNP Synthesis

Solution A: 1% (*w*/*v*) HAuCl_4_·3H_2_O (1.16 mL) and deionized water (79 mL). Solution B: 1% sodium citrate (4.5 mL), 1% tannic acid (0.025 mL) and deionized water (15.5 mL). Solutions A and B were heated separately to 60 °C under magnetic stirring in a round bottom flask. Both solutions were mixed (B is added to A). The temperature was increased to 90 °C. After color change (yellow to red), the temperature of the bath increased to 120 °C and the mixture was stirred for 30 min after first bubbling. After synthesis, nanoparticles were cooled down in an ice bath. The colloidal solution of AuNPs was characterized by UV–visible spectroscopy, Dynamic Light Scattering (DLS), zeta potential and Transmission Electron Microscopy (TEM).

### 2.7. Conjugation of Goat Anti-rIgG to AuNP

Conjugation of goat anti-rIgG to AuNP was carried out according to the previously reported chemisorption method [29] with slight modifications. Thiolation of goat anti-rIgG antibody was performed by adding Traut’s reagent in phosphate buffer pH 8 (10 mM; 6.6 µL; 20 eq.) to Ab (1 mg/mL in phosphate buffer pH 8; 500 μL). After 1 h, excess reagent was separated from the antibody by gel filtration on a desalting column (10 mL; ThermoFisher Sci.) with 10 mM phosphate buffer pH 7.4 (PB) as eluent. Twelve fractions (1 mL) were collected and absorbance measured in 280 nm. Fractions containing Ab-SH were pooled and immediately used for conjugation to AuNP. AuNP solution (2.8 nM; 5 mL in PB) was mixed with Ab-SH (0.21 mg/mL; 250 μL). After 12 h, solid BSA was added to the suspension to a final concentration of 0.25%. After another 10 min, AuNP were washed twice with PB (13,000 rpm, 30 min). The final pellet was dispersed in the same buffer supplemented with 0.1% BSA (5 mL).

### 2.8. Determination of Number of Ab Conjugated to AuNP

Determination of antibody coverage on AuNP was performed according to a previously reported indirect quantification method [28] using goat anti-rIgG labeled with FITC with slight changes. Thiolation of FITC-labeled antibody and conjugation to AuNP were performed according to the methods described in 2.7., except that AuNP-Ab conjugate was washed four times with PB (13,000 rpm, 30 min). The fourth pellet was dispersed in PB-0.25% BSA. The four supernatants were kept for spectrofluorimetric analysis to quantify the amount of unbound FITC-labeled Ab.

### 2.9. Magneto-Plasmonic Immunosensor

MB-Ab suspension (3.8 mg/mL; 10 μL) was dispensed in microtubes, collected by a magnet and supernatants discarded. MB-Ab was washed with PB. Rabbit IgG in PB (1 ng/mL to 10 µg/mL; 50 μL) was immediately added. As a negative control, rIgG was replaced by PB. The suspensions were incubated for 2 h under rotative stirring. Then, magnetic beads were collected by a magnet and washed twice with PB to remove unbound rabbit IgG. AuNP-Ab conjugate (3.8 nM; 50 μL) was added to the magnetic particles and further incubated overnight. Immunocomplexes were collected with a magnet and supernatants transferred to microtubes. PB (50 μL) was added to each supernatant to a final volume of 100 μL and used for the indirect detection. Magnetic beads were dispersed in PB (100 μL) and kept for the direct detection.

### 2.10. Characterization Methods and Optical Measurements

The AuNP and magnetic particles were analyzed on a Cary 50 spectrophotometer (Varian Inc., Palo Alto, CA, USA) in the range 300–800 nm. Dynamic light scattering (DLS), and zeta potential (ELS) measurements were performed on Litesizer™ 500 apparatus (Anton Paar, Graz, Austria) equipped with a 658 nm laser operating at 40 mW. We set the backscattered light collection angle to 175°. The zeta potential was measured in a Ω-shaped capillary tube cuvette, the applied potential was of 150 V. Transmission electron microscopy (TEM) analysis of AuNP and MB was performed using a JEM 1011 plus LaB6 microscope (JEOL; Croissy-sur-Seine, France) operating at an accelerating voltage of 200 kV and equipped with an Orius 4 K CDD camera using a carbon-coated copper grid. Scanning electron microscopy (SEM) was performed using a FEG SU-70 scanning electron microscope (Hitachi, Krefeld, Germany) with free field mode to avoid possible contamination of the electromagnetic lens with the magneto plasmonic complex. The different images were recorded with accelerated electron beam at 15 kV and a magnification of 80,000× at working distance of 10 mm from the specimen; the secondary electron detector “in Lens” was used. The size distribution of AuNP was determined using Image J. Extinction spectra of immunocomplexes and supernatants were measured in the microscale or the microfluidic cell on XNano instrument (Insplorion, Göteborg, Sweden).

## 3. Results

### 3.1. Synthesis and Characterization of AuNP and Bioconjugation of Antibody

AuNP were synthesized by reduction of HAuCl_4_ with citrate following the Turkevich method [30] and characterized by UV–visible spectroscopy, TEM, DLS and zeta potential measurements (Figure 1 and Table 1).

The extinction spectrum of AuNP shows the expected typical plasmon peak at 520 nm (Figure 1e). DLS and ELS characterizations gave a mean hydrodynamic diameter (intensity weighted peak) of 18.65 nm with a polydispersity index of 0.16 and a mean zeta potential of −59 mV (Figure 1c and Table 1). According to the TEM images (Figure 1a and Table 1), the AuNP are spherical and have a narrow distribution in size and shape with an average diameter of 15 nm. The concentration of AuNP solution was estimated to be 2.77 nM using an extinction coefficient of 3.8 × 10^8^ M^−1^·cm^−1^ taken from the literature [31].

Goat anti-rabbit IgG antibody was first thiolated with Traut’s reagent (iminothiolane) then conjugated to AuNP by formation of Au-S bonds (Figure 2) according to a previously described procedure [29].

The resulting AuNP-Ab conjugate was characterized by DLS, ELS, UV–visible spectroscopy and TEM (Figure 1 and Table 1). As expected, the LSPR band of AuNP red-shifted from 520 to 529 nm due to Ab attachment (Figure 1e). In addition, the UV–visible spectrum of AuNP-Ab displayed another absorption band at 280 nm, mostly resulting from the presence of a high concentration of BSA in the storage buffer (Figure 1e). Assuming the same extinction coefficient of AuNP, the concentration of AuNP-Ab solution was equal to 1.6 nM. TEM gave an average diameter of AuNP-Ab centered at 16.2 ± 1.2 nm. Moreover, the mean hydrodynamic diameter (intensity weighted peak) increased to 103 nm (polydispersity index = 0.20), and the mean zeta potential of antibody-conjugated AuNP increased to −15 mV in agreement with the literature [9,28] (Figure 1c and Table 1).

The surface coverage of anti-rIgG Ab on AuNP was determined according to our previously reported indirect procedure relying on the chemisorption of FITC-labeled Ab to AuNP followed by the assay of unbound antibody by spectrofluorimetry (Appendix A) [28]. On average, 14 Ab molecules were grafted per AuNP. This value is in good agreement with previous work [28] and gives an acceptable estimation of the number of antibodies immobilized on AuNP via chemisorption.

### 3.2. Conjugation of Antibody to Magnetic Beads

Goat anti-rabbit IgG antibody was conjugated to NHS-activated magnetic beads (MB) via their amine functions according to the manufacturer’s instructions (Figure 2). The physical characterizations of magnetic beads before and after conjugation of Ab are shown in Figure 3 and Table 2.

SEM and TEM images of MB show that the beads display an irregular shape with a mean diameter around 1 µm (Figure 3a,b). Covalent attachment of Ab to MB was validated by a colorimetric ELISA test using MB-BSA as a negative control (Appendix A). Conjugation of the antibody induced an increase of the mean hydrodynamic diameter of the beads from 1.11 µm to 1.24 µm as measured by DLS and a slight increase of the zeta potential from −23 to −17 mV (Figure 3c and Table 2). These data provide additional evidence of antibody conjugation to the magnetic beads. MB and MB-Ab display a broad absorption in the whole visible spectral range that partially overlaps the LSPR band of AuNP-Ab, as can be seen in Figure 3e.

### 3.3. Magneto-Plasmonic Immunodetection of Rabbit IgG

#### 3.3.1. Characterization of Magneto-Plasmonic Immunocomplex

Immunodetection of rabbit IgG by the magneto-plasmonic immunosensor consists of a sandwich assay where the analyte is first mixed with a fixed amount of MB-Ab allowing its specific capture. After elimination of unbound analyte, a fixed amount of AuNP-Ab conjugate is added and binds to the antigen captured by MB-Ab to form a sandwich immunocomplex. Application of an external magnetic field allows to separate the magneto-plasmonic immunocomplex fraction (solid phase) from the unbound AuNP-Ab fraction (supernatant) that are separately injected in a home-made microcell and analyzed in the XNano instrument (Figure 4). Such a configuration was reported earlier for the immunoassay of a protein biomarker [22] and a mycotoxin [24]. Let us note, however, that in both cases, only the unbound fraction of the AuNP-Ab immunoprobe was analyzed because of the high absorption of the magnetic beads.

In a preliminary experiment, the magneto-plasmonic immunoassay was performed with a 1 µg/mL solution of rabbit IgG. The magnetic beads’ fractions obtained after the last magnetic separation were analyzed by TEM and SEM (Figure 5a–c). Both TEM and SEM images of a single MB clearly showed the presence of 16 nm-diameter AuNP at different spots on the surface of the MB in agreement with the formation of the immunocomplex.

The immunocomplex fraction produced from rabbit IgG at 1 µg/mL was also analyzed by UV–vis spectroscopy using a specially designed microscale cell. Before incubation of the rabbit IgG and AuNP-Ab, the absorption spectrum shows an extremely broad absorption band whose maximum is located at 453 nm (Figure 5d). Addition of the rabbit IgG and AuNP-Ab probe results in a significant change of the absorption spectrum with the appearance of the plasmonic signature of the AuNP between 500 and 600 nm overlapping the absorption of the MB. To extract the plasmonic response from the extinction spectra, first order derivatives of both extinction spectra were calculated (Figure 5d). The contribution of the plasmonic signature of the bound AuNP-Ab clearly appeared as a broad band between 475 and 585 nm (blue zone in Figure 5d). Therefore, the integrated area of this band will be used to quantify the amount of the bound fraction of the AuNP-Ab probe (See Appendix A).

#### 3.3.2. Direct Detection of Rabbit IgG with Magneto-Plasmonic Immunosensor

Magneto-plasmonic complexes were formed by addition of variable concentrations of rabbit IgG (0–10.4 µg/mL). Extinction spectra of suspensions of immunocomplexes were recorded (Figure 6a) and the dose–response curve was established by plotting the integrated area between 475 and 585 nm versus rIgG concentration (Figure 6b).

Curve fitting of the data was performed by applying the Langmuir isotherm equation (Equation (1)):(1)$$\;\theta =\frac{K_{eq}\times C}{1+K_{eq}\times C}$$
where *θ* is the fraction of surface covered by adsorbate; *K_eq_* is the equilibrium constant of adsorption; and *C* is the concentration of absorbate. Three critical assumptions for Langmuir adsorption isotherms equation include: (a) The adsorbate covers the surface up to complete coverage as a monolayer on the substrate; (b) There are no adsorbate–adsorbate interactions on the surface of the host substrate; (c) all binding sites are equivalent.

The Langmuir adsorption isotherm Equation (1) was rearranged as follows (Equation (2)):(2)$$A=\frac{\left[rIgG\right]\times A_{max}}{K_{d}+\left[rIgG\right]}$$
where *A* refers to the integrated area of the LSPR band of the sandwich immunocomplex measured on the first order derivative spectra; *K_d_* refers to the dissociation constant between AuNP-Ab bioconjugate and rabbit IgG and shows the affinity of AuNP-Ab conjugate for rabbit IgG; and [*rIgG*] represents the concentration of rabbit IgG.

*A_max_* and *K_d_* were obtained by fitting the dose–response curve to Equation (2) (Figure 6, Table 3). The calculated value for *K_d_* is 430 ± 170 ng/mL. The limit of detection of rabbit IgG calculated from 3.3 times the standard deviation of the blank was 70 ng rIgG/mL.

#### 3.3.3. Indirect Detection of Rabbit IgG

To bypass the strong absorption of magnetic particles in the visible range, an alternative strategy is to analyze the supernatants containing the unbound fraction of AuNP-Ab conjugate remaining after the immunoreaction. Figure 7a shows the absorption spectra of unbound AuNP-Ab after separation of the immunocomplexes for different concentrations of rabbit IgG and Figure 7b shows the dose–response curve obtained by plotting the intensity of the plasmonic band at 530 nm as a function of rabbit IgG concentration. As could have been expected, the response is now inversely related to the concentration of rabbit IgG and for the highest concentration of rIgG, the plasmonic signal was below the detection limit of the instrument, indicating that most of the added AuNP-Ab immunoprobe was bound to the MB-Ab.

Data points were fitted according to the 4-parameter logistic equation (Equation (3)):(3)$$y=y_{min}+\frac{y_{max}-y_{min}}{1+{\left(\frac{x}{IC_{50}}\right)}^{HS}}$$

The fitting parameters are shown in Table 3 as well as the analytical performances of the magneto-plasmonic immunosensor in the direct and indirect formats. The mid-point (*IC*_50_) of the calibration curve was equal to 0.42 ± 0.06 µg rIgG/mL and the limit of detection of rabbit IgG, calculated from 3.3 times the standard deviation of the blank, was equal to 0.23 µg/mL.

#### 3.3.4. Indirect Detection of Rabbit IgG with Microfluidic Cell

To go further towards the development of a self-contained setup, we designed a microfluidic cell that integrates the magnetic separation and readout steps in a single device. The microfluidic cell designed for this study was composed of four parts labeled in blue in Figure 8: inlet and outlet connected with PTFE tubing, a serpentine channel, and a readout window. The function of the serpentine channel is to control the volume of solution directly exposed to the magnetic field of the magnet. The section of the channel is a truncated disk (h/r: 0.3/0.4 mm). The volume of the serpentine channel is 11.6 µL. The volume of the 3− mm diameter readout window was set to 17.7 µL to be able to dump all the serpentine volume in the readout.

In a typical experiment, an empty cell was inserted in the Insplorion XNano sample holder with a cylindrical magnet mounted over the serpentine channel. A continuous UV–vis acquisition was launched, where the blank was recorded with an empty cell. Different syringes were loaded with 100 µL of sample with rabbit IgG concentration varying between 0.001 and 2 µg/mL and containing both the immunocomplex and free AuNP-Ab. Each syringe was connected to the cell using a syringe pump applying a flow rate of 15 µL/min. The immunocomplex was instantly trapped by the magnet while free AuNP-Ab accumulated in the readout window. Once the UV–vis spectrum from the readout was stable, it was exported and the cell was rinsed with Milli-Q water and dried. A dose–response curve was constructed by plotting the intensity of the LSPR band as a function of analyte concentration and data points were fitted to the 4-parameter logistic Equation (3) (Figure 9).

The fit parameters and analytical performances of the magneto-plasmonic immunosensor in the indirect format using the microfluidic cell are gathered in Table 3. They show the same trend as that observed with the microscale cell for the indirect readout. However, interestingly, the LoD was found much lower in the microfluidic cell, 1.2 ng/mL vs. 230 ng/mL in the microscale cell. This promising LoD needs to be confirmed and further investigated for other targets. Our ongoing studies aim at understanding its origin and applying this magneto-plasmonic immunosensor to the detection of various targets related to health and/or food security.

## 4. Conclusions

We have successfully developed an enzyme-free, sandwich immunomagnetic assay using gold nanoparticle-based colorimetric probe for the optical detection of antigens. This assay combines antibody-conjugated magnetic beads having a high surface to volume ratio for the efficient capture of the analyte and easy separation with a magnet and antibody-conjugated gold nanoparticles having extremely high absorption properties in the visible spectral range. By using appropriate mathematical treatment of the raw extinction spectra, both the bound and the free fractions of AuNP-Ab were measured from which dose–response curves could be derived. Both calibration curves displayed similar mid-point concentrations but the LoD was ca. three times lower for the direct detection. Transposing the assay to a microfluidic chip enabled a dramatic decrease in the assay sensitivity with a LoD in the ng/mL range. This immunosensor could be readily translated to more biological meaningful biomarkers related to health and/or food security.

## Figures and Tables

**Figure 1 biosensors-12-00799-f001:**
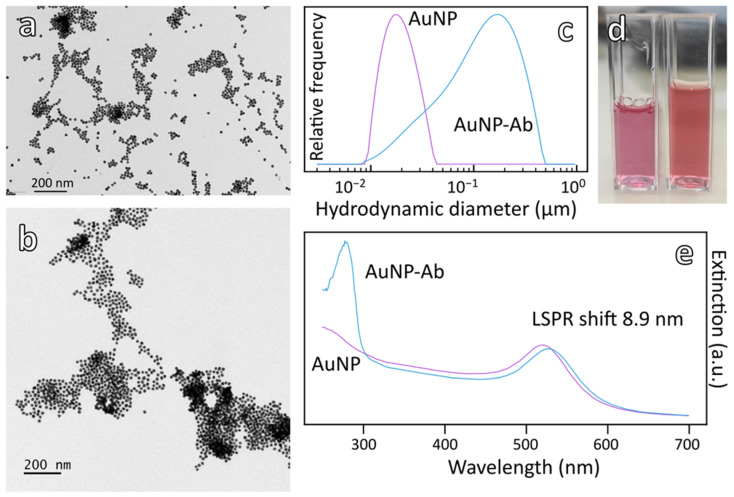
Physical characterizations of AuNP and AuNP-Ab conjugate. (**a**) TEM image (10,000×) before (AuNP) and (**b**) after antibody (Ab) conjugation (AuNP-Ab) (12,000×), (**c**) DLS measurement before (purple) and after (blue) antibody conjugation to AuNP, (**d**) photograph of visual color difference, before (right, red) and after (left, purple) antibody conjugation to AuNP with corresponding, (**e**) UV–vis spectra before (purple) and after (blue) Ab conjugation.

**Figure 2 biosensors-12-00799-f002:**
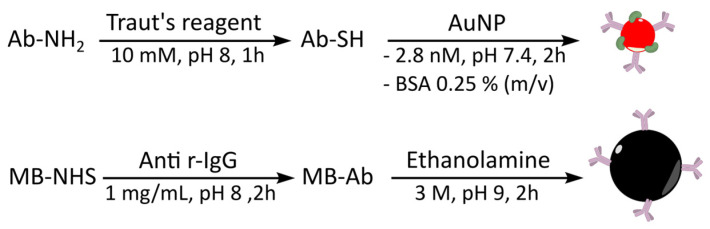
Conjugation of goat anti-rabbit IgG (Ab) to AuNP and NHS-activated magnetic beads (MB).

**Figure 3 biosensors-12-00799-f003:**
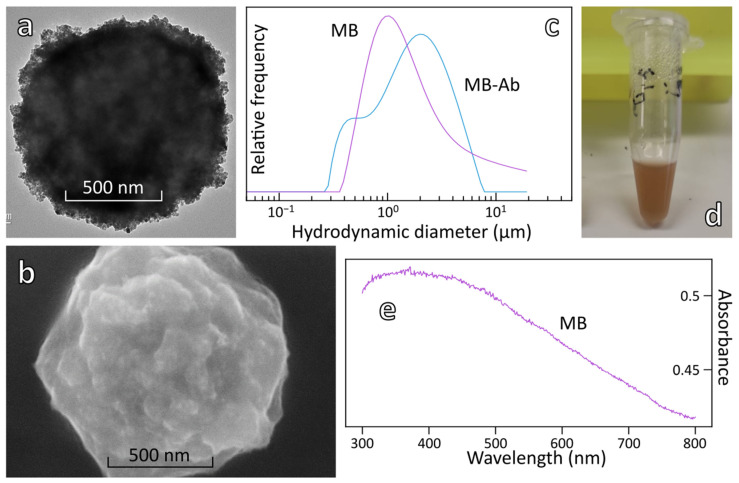
Physical characterizations of magnetic beads before and after conjugation of Ab, (**a**) TEM image (15,000×) and (**b**) SEM image of MB-Ab conjugate, (15 kV, 10.3 mm, 80,000×), (**c**) DLS measurements before (purple) and after (blue) conjugation of MB with antibodies, (**d**) Photograph of magnetic beads suspension with corresponding (**e**) UV–vis spectrum.

**Figure 4 biosensors-12-00799-f004:**
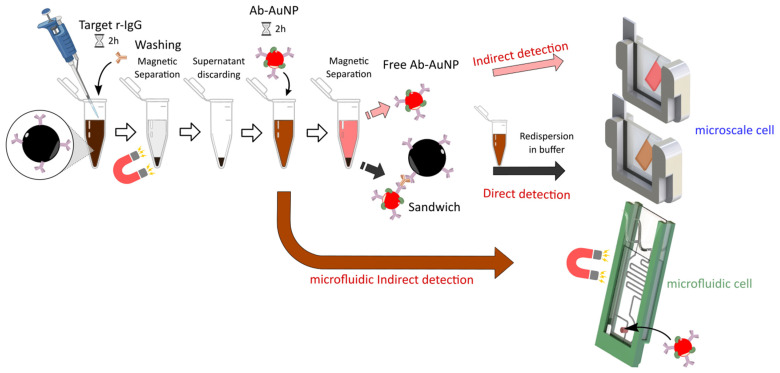
Principle of magneto-plasmonic immunoassay of rabbit IgG.

**Figure 5 biosensors-12-00799-f005:**
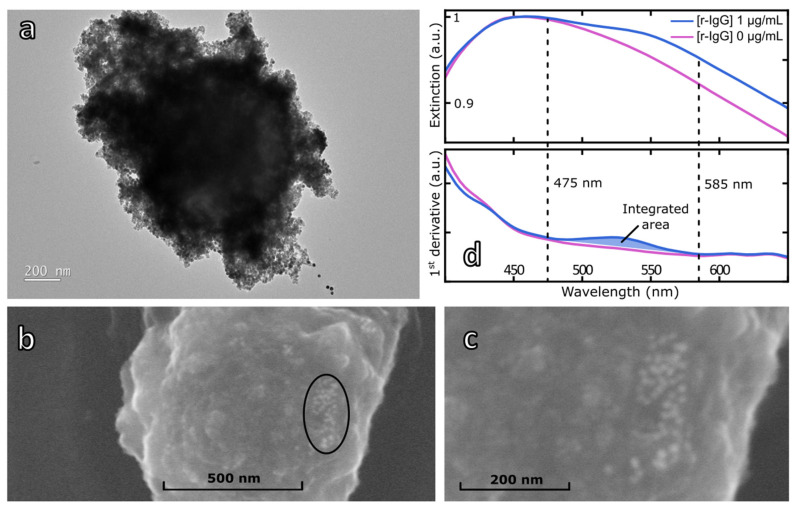
Physical characterizations of magneto-plasmonic immunocomplex. (**a**) TEM image (15,000×) and (**b**) SEM image (15 kV, 10.3 mm, 80,000×) of the magneto-plasmonic immunocomplex in the presence of 1 µg/mL of rIgG and a black circle showing the presence of AuNPs. (**c**) Zoom on the AuNPs. (**d**) Normalized and smoothed extinction spectra (**top**) and first order derivatives (**bottom**) of MB (pink trace) and immunocomplex resulting from the incubation of 1 µg/mL rabbit IgG and AuNP-Ab (blue trace).

**Figure 6 biosensors-12-00799-f006:**
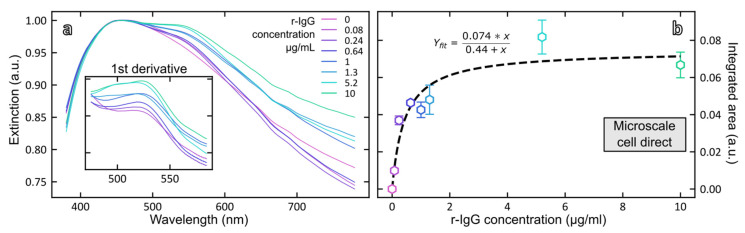
(**a**) Normalized and smoothed extinction spectra of immunocomplex suspensions resulting from the addition of rabbit IgG in the range 0–10 µg/mL; (**b**) Dose–response curve plotted from the integrated area of the first order derivative spectra between 475 and 585 nm vs. rabbit IgG concentration.

**Figure 7 biosensors-12-00799-f007:**
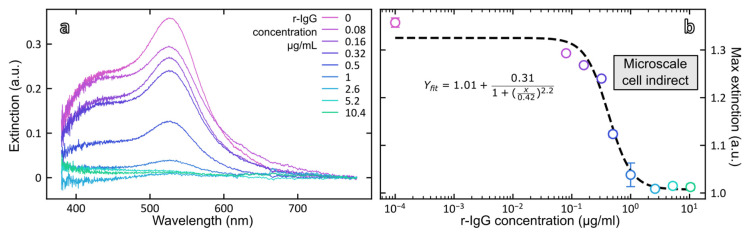
(**a**) Extinction spectra of the unbound fraction of AuNP-Ab after incubation of rabbit IgG in the range 0–10.4 µg/mL measured in the microcell; (**b**) Dose–response curve plotted from the maximum extinction at 530 nm vs. rabbit IgG concentration.

**Figure 8 biosensors-12-00799-f008:**
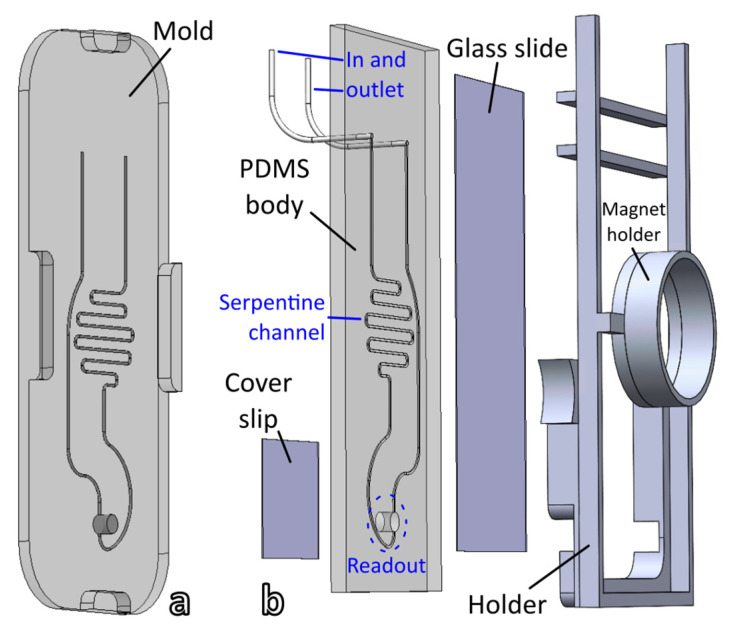
(**a**) Schematic of the 3D-printed mold utilized to shape the PDMS microfluidic cell. The mold is placed face down onto viscous PDMS, the four shoulders determine the thickness of the cell. (**b**) Exploded view of the four components of the microfluidic cell. The glass cover slip and slide are glued to the PDMS using an oxygen plasma. The assembly is then mounted on the cartridge holder to perform UV–vis spectrometry.

**Figure 9 biosensors-12-00799-f009:**
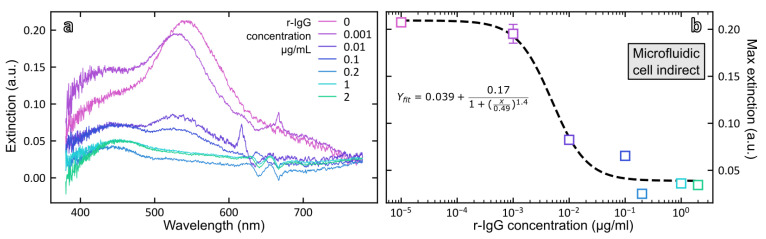
(**a**) Extinction spectra of the unbound fraction of AuNP-Ab after incubation of rabbit IgG in the range 0–2 µg/mL measured in the microfluidic cell; (**b**) Dose–response curve plotted from the maximum extinction at 530 nm vs. rabbit IgG concentration.

**Table 1 biosensors-12-00799-t001:** Hydrodynamic diameter (HD), polydispersity (PD), zeta potential and TEM size distribution for bare (AuNP) and conjugated (AuNP-Ab).

	HD (nm)	PD (%)	Zeta Pot. (mV)	TEM Size (nm)
AuNP	18.6	16.5	−59.2 ± 0.9	15.3 ± 1.5
AuNP-Ab	105.3	28.2	−19.1 ± 0.6	16.2 ± 1.2

**Table 2 biosensors-12-00799-t002:** Hydrodynamic diameter (HD), polydispersity (PD), zeta potential and TEM size distribution for bare (MB) and conjugated (MB-Ab).

	HD (µm)	Zeta Pot. (mV)	TEM Size (µm)
MB	1.11	−23.3 ± 0.5	1.1 ± 0.1
MB-Ab	1.24	−17.3 ± 1.1	

**Table 3 biosensors-12-00799-t003:** Fitting parameters and analytical performances of the magneto-plasmonic immunosensor in the direct and indirect formats.

	Microscale Cell	Microfluidic Cell
Indirect detection	χ^2^ = 0.00065	χ^2^ = 0.00028
	R^2^ = 0.979	R^2^ = 0.976
	*IC*_50_ = 0.42 ± 0.06 µg/mL	*IC*_50_ = 4.9 ± 2.2 ng/mL
	LoD = 230 ng/mL ^a^	LoD = 1.2 ng/mL ^b^
Direct detection	χ^2^ = 0.000076	Not done
	R^2^ = 0.91	
	*K_d_* = 0.43 ± 0.17 µg/mL	
	LoD = 70 ng/mL ^a^	

^a^ calculated from 3.3 times the standard deviation of the blank; ^b^ calculated from 90% of the maximum response.

## Data Availability

Data is contained within the article and Appendix A.

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
