# Peer review of "Design and Optimization of A Magneto-Plasmonic Sandwich Biosensor for Integration within Microfluidic Devices"

_biosensors, 2022, doi:10.3390/bios12100799_

Round 1
Reviewer 1 Report
Comments:
Although there are many methods using Au NP and MP for detecting the antigens, Mona and co-workers designed a magneto-plasmonic biosensor for the immunodetection of antigens in minute sample volume, which is interesting.
Mona and co-workers designed a magneto-plasmonic biosensor to the immunodetection of antigens in miner sample volume. AuNP and magnetic beads were conjugated with goat-anti-rIgG antibody. AuNP with antibody was used as optical detection probe while MB was used as capture probe. Once the target analyte were added, the detection probe will form a sandwich immunocoplex which can be removed by external magnet. There are many methods have been developed to optical detect the antigens using this sandwich immunocoplex through binding of AuNP and magnetic nanoparticles. However, the author showed in this manuscript that this systems can detect antigens in minor sample volume and finally they transposed to a microfluidic cell and achieve the quantification of the analyte in the ng/mL concentration range. Overall, this is an interesting work suitable for Biosensor. Some of minor issues need to be addressed before being accepted for publication.
1, In the line 197, 225, there are some error happened, there are text like“(Erreur ! Source du renvoi introuvable.) ”. The authors should check them.
2, There are two Figure 1. The authors should make correct Figure number and cite in the right place. It’s better to cite each figures under the text description. “The extinction spectrum of AuNP shows the expected typical plasmon peak at 520 nm. DLS and ELS characterizations gave a mean hydrodynamic diameter (intensity weighted peak) of 18.65 nm with a poly dispersity index of 0.16 and a mean zeta potential of -59 mV. According to the TEM images, AuNP are spherical and have a narrow distribution in size and shape with an average diameter of 15 nm.” And also correct and cire the right Figures here “As expected, the LSPR band of AuNP red-shifted from 520 to 529 nm due to Ab attachment. In addition, the UV-Visible spectrum of AuNP-Ab displayed another absorption band at 280 nm, mostly resulting from the presence of a high concentration of BSA in the storage buffer. Assuming the same extinction coefficient of AuNP, the concentration of AuNP-Ab solution was equal to 1.6 nM. TEM gave an average diameter of AuNP-Ab centered at 16.2 ± 1.2 nm.”
3, The author should show the number of Table in Figures 1 and 2 separate from Figure 1E and cite the right place.
4, The author shows that high absorption band at 280 nm, it was caused by high concentration of BSA in the storage buffer. The author should explain why BSA was necessary in the storage buffer. Whether it can be removed before measurement and see whether it affect that absorption.
5, The author write in the manuscript that “14 Ab molecules were grafted per AuNP” The authors should explain the calculate method for the AB molecules on the surface of AuNP.
6, The authors should show the average value with error bar for measuring the Rabbit IgG conjugated magnetic beads in Figure S2. In addition, From the results, the high concentration showed high conjugate of antibody. The author should clear descript what concentration is used in the process of detection.
7, The author should cite the Figure number clearly, like separately and clearly cite Figure 4a, b c and d at line 278-298. Double check other Figures, It’s better to cite Figures separately and clearly.
8, The author should cite the Table 1 at ling 338, and 387 or somewhere it fitted.
Author Response
Thank you very much for your positive evaluation of our work, below a detailed response to each point.
1, In the line 197, 225, there are some error happened, there are text like“(Erreur ! Source du renvoi introuvable.) ”. The authors should check them.
We apologize for this error that happened upon formatting and have corrected it all through the manuscript.
2, There are two Figure 1. The authors should make correct Figure number and cite in the right place. It’s better to cite each figures under the text description. “The extinction spectrum of AuNP shows the expected typical plasmon peak at 520 nm. DLS and ELS characterizations gave a mean hydrodynamic diameter (intensity weighted peak) of 18.65 nm with a poly dispersity index of 0.16 and a mean zeta potential of -59 mV. According to the TEM images, AuNP are spherical and have a narrow distribution in size and shape with an average diameter of 15 nm.” And also correct and cire the right Figures here “As expected, the LSPR band of AuNP red-shifted from 520 to 529 nm due to Ab attachment. In addition, the UV-Visible spectrum of AuNP-Ab displayed another absorption band at 280 nm, mostly resulting from the presence of a high concentration of BSA in the storage buffer. Assuming the same extinction coefficient of AuNP, the concentration of AuNP-Ab solution was equal to 1.6 nM. TEM gave an average diameter of AuNP-Ab centered at 16.2 ± 1.2 nm.”
Again we apologize for this error that happened upon formatting and have corrected it through the manuscript. In addition we have modified the citation to the figures following the reviewers recommendations
3, The author should show the number of Table in Figures 1 and 2 separate from Figure 1E and cite the right place.
We have modified the figures accordingly and tables accordingly. We have also merged the two first paragraphs to better refer to the figures and tables.
- The author shows that high absorption band at 280 nm, it was caused by high concentration of BSA in the storage buffer. The author should explain why BSA was necessary in the storage buffer. Whether it can be removed before measurement and see whether it affect that absorption.
Addition of 0.1% BSA in the storage buffer of gold nanoparticles conjugated to proteins is a classical method to maintain the long term stability of the colloidal solutions, see for instance the recommendations given by G.T. Hermanson in Bioconjugate Techniques, chapter 14; Academic press, pp. 593-604.
5, The author write in the manuscript that “14 Ab molecules were grafted per AuNP” The authors should explain the calculate method for the AB molecules on the surface of AuNP.
We clarified the determination of the antibody-to-AuNP ratio in the supplementary information in the form of a paragraph above Fig. S1 including experimental and calculation details.
6, The authors should show the average value with error bar for measuring the Rabbit IgG conjugated magnetic beads in Figure S2. In addition, From the results, the high concentration showed high conjugate of antibody. The author should clear descript what concentration is used in the process of detection.
Error bars were added to the histogram included in Fig. S2. The concentration of antibody used for the grafting experiments is indicated below the x-axis of the histogram.
7, The author should cite the Figure number clearly, like separately and clearly cite Figure 4a, b c and d at line 278-298. Double check other Figures, It’s better to cite Figures separately and clearly.
All the figures’ citations were modified accordingly
8, The author should cite the Table 1 at ling 338, and 387 or somewhere it fitted.
Modified as suggested
Reviewer 2 Report
referee report
biosensors-1922020-peer-review-v1
Design and optimization of a magneto-plasmonic sandwich bi-2 osensor for integration within microfluidic devices
Mona Soroush et al.
The present article describes the design and the test results of a magneto-plasmonic biosensor for the immunodetection
of antigens. Spherical gold nanoparticles (AuNP, for optical detection) and magnetic beads (MB, for capturing) were
employed as probes, allowing the measurements of the UV-visible spectrum. The development of such biosensors is very
interesting for various applications in medicine and fields like food control or environmental monitoring. Thus, the
topic is well suited for Biosensors.
The manuscript comprises 8 figures, 1 table and 31 references are given, which provide a good overview on this research
field. Furthermore, three additional figures (S1--S3) are provided as Supplementary material.
Overall, the manuscript is well arranged, and a detailed description of all experimental steps involved is given in Sec. 2
(2.1.--2.10.). The English of the manuscript is quite well, except some minor points and the use of commas.
All figures are well prepared including the necessary scale bars in the SEM and TEM images. Thus, the manuscript makes
a proper impression at the first glance. However, when looking more in detail into the text, there are several problems
which must be corrected prior to publication. These points are listed below:
# On page 5 and page 6, there is the comment: (Erreur ! Source du renvoi introuvable.) Obviously, a required reference
did not function properly.
# Formulae and the mathematics are not written in an unified manner throughout the manuscript. Such different styles
should be avoided. Example line 37: dot at the bottom line for "times" vs. line 80: dot at the center line for "times".
Please unify this; anyway, the dot can be skipped.
# There should always be a space between a physical quantity and its unit. This is done well in many places in the
manuscript, but there are also many places without this space.
# Is the captial M denoting the molar mass or the mol?
# Figure 8a is mentioned before any other figure. Should this be then Figure 1?
# line 100: 24 x 60 glass slide: Now, the "x" is used as multiplicator, and the unit is missing.
# line 103: Torr is NOT an SI unit.
# line 134/182: chemical formula
# line 208/Fig.1: A magnification is normally written as, e.g., 10000x
# line 253 should read: 15 kV, 10.3 mm, 80000x. Same for Fig. 4.
# Please also use a common style for all mathematical expressions. Your formula writing program is using italics for
all physical quantities, which is correct. Please also do it like this in the normal text. Any subscripts -- if
they do not belong the unit itself, should be written in roman, like you do it in the text.
To sum up, the present manuscript provides intereting material, well worth for publication. However, all the points
mentioned should be corrected.
Author Response
Thank you very much for your positive evaluation of our work, below a detailed response to each point.
# On page 5 and page 6, there is the comment: (Erreur ! Source du renvoi introuvable.) Obviously, a required reference did not function properly.
We apologize for this error that happened upon formatting and have corrected it all through the manuscript
# Formulae and the mathematics are not written in an unified manner throughout the manuscript. Such different styles
should be avoided. Example line 37: dot at the bottom line for "times" vs. line 80: dot at the center line for "times".
Please unify this; anyway, the dot can be skipped.
We have modified and unified all the formula in the manuscript
# There should always be a space between a physical quantity and its unit. This is done well in many places in the manuscript, but there are also many places without this space.
Thank you, this was modified accordingly
# Is the captial M denoting the molar mass or the mol?
Yes the capital M refers to mol.L-1, to clarify we changed it in the manuscript
# Figure 8a is mentioned before any other figure. Should this be then Figure 1?
We do apologize for this mistake, the reference to Figure 8 was removed from the experimental part
# line 100: 24 x 60 glass slide: Now, the "x" is used as multiplicator, and the unit is missing.
Thank you, we have added the unit and checked everywhere in the manuscript that we are using the mathematical multiplier ×
# line 103: Torr is NOT an SI unit.
Agreed, Torr was replaced by Pa
# line 134/182: chemical formula
Thank you, corrected
# line 208/Fig.1: A magnification is normally written as, e.g., 10000x
Agreed, modified accordingly
# line 253 should read: 15 kV, 10.3 mm, 80000x. Same for Fig. 4.
Agreed, modified accordingly
# Please also use a common style for all mathematical expressions. Your formula writing program is using italics for
all physical quantities, which is correct. Please also do it like this in the normal text. Any subscripts -- if
they do not belong the unit itself, should be written in roman, like you do it in the text.
Agreed, modified accordingly